# Intraamniotic Zika virus inoculation of pregnant rhesus macaques produces fetal neurologic disease

Lark L. Coffey [1], Rebekah I. Keesler[2], Patricia A. Pesavento[1], Kevin Woolard[1], Anil Singapuri[1], Jennifer Watanabe[2], Christina Cruzen[2], Kari L. Christe[2], Jodie Usachenko[2], JoAnn Yee[2], Victoria A. Heng[2,3], Eliza Bliss-Moreau[2,4], J. Rachel Reader [2], Wilhelm von Morgenland[2], Anne M. Gibbons[2], Kenneth Jackson[1], Amir Ardeshir [2], Holly Heimsath[5], Sallie Permar[5], Paranthaman Senthamaraikannan [6], Pietro Presicce[7], Suhas G. Kallapur[7], Jeffrey M. Linnen[8], Kui Gao[8], Robert Orr[9], Tracy MacGill[9], Michelle McClure[10], Richard McFarland[11], John H. Morrison[2] & Koen K.A. Van Rompay [2]

Zika virus (ZIKV) infection of pregnant women can cause fetal microcephaly and other neurologic defects. We describe the development of a non-human primate model to better understand fetal pathogenesis. To reliably induce fetal infection at defined times, four pregnant rhesus macaques are inoculated intravenously and intraamniotically with ZIKV at gestational day (GD) 41, 50, 64, or 90, corresponding to first and second trimester of gestation. The GD41-inoculated animal, experiencing fetal death 7 days later, has high virus levels in fetal and placental tissues, implicating ZIKV as cause of death. The other three fetuses are carried to near term and euthanized; while none display gross microcephaly, all show ZIKV RNA in many tissues, especially in the brain, which exhibits calcifications and reduced neural precursor cells. Given that this model consistently recapitulates neurologic defects of human congenital Zika syndrome, it is highly relevant to unravel determinants of fetal neuropathogenesis and to explore interventions.

[1] Department of Pathology, Microbiology and Immunology, School of Veterinary Medicine, University of California, 1 Shields Avenue, Davis, CA 95616, USA. [2] California National Primate Research Center, University of California, 1 Shields Avenue, Davis, CA 95616, USA. [3] Donders Institute, Radboud University, Montessorilaan 3, 6525 HR Nijmegen, The Netherlands. [4] Department of Psychology, University of California, 1 Shields Avenue, Davis, CA 95616, USA. [5] Duke Human Vaccine Institute, Duke University Medical Center, 103020, 2 Genome Court MSRBII, Durham, NC 27710, USA. [6] Division of Neonatology and Pulmonary Biology, Cincinnati Children's Hospital Research Foundation, 3333 Burnet Avenue, Cincinnati, OH 45229, USA. [7] Divisions of Neonatology and Developmental Biology, David Geffen School of Medicine at the University of California, 10833 Le Conte Avenue, Los Angeles, CA 90095, USA. [8] Grifols Diagnostic Solutions, Inc., 10808 Willow Court, San Diego, CA 92127, USA. [9] Office of Counterterrorism and Emerging Threats, Office of the Chief Scientist, Food and Drug Administration, 25 New Hampshire Avenue, Silver Spring, MD 20903, USA. [10] Office of Tissues and Advanced Therapies, Center for Biologics Evaluation and Research, Food and Drug Administration, 10903 New Hampshire Avenue, Silver Spring, MD 20903, USA. [11] The Advanced Regenerative Manufacturing Institute, 400 Commercial Street, Manchester, NH 03101, USA. Correspondence and requests for materials should be addressed to L.L.C. (email: lcoffey@ucdavis.edu) or to K.K.A.V.R. (email: kkvanrompay@ucdavis.edu)

Fetuses from pregnant women infected with Zika virus (ZIKV) sometimes display reduced growth, arthrogryposis, ocular calcifications, skeletal and sensory disorders, and central nervous system (CNS) malformations including calcifications and aberrant neural cell development[1–8], manifestations that are together termed congenital Zika syndrome[9–11]. Animal models of human ZIKV infection in pregnancy are essential for determining pathogenesis and long-term effects on the infant brain, as well as treatments or vaccines to control or prevent infection and teratogenic effects in pregnant women. Although ZIKV-infected mice show fetal growth restriction, brain infection, and in some systems, CNS lesions similar to those seen in humans[12–19], significant differences in CNS development between mice and humans remain as limitation of murine models. In contrast, due to more similar placentation, immunology, fetal organogenesis, and neurologic development, macaques are an emerging model for understanding ZIKV infection and disease[20–29], and testing candidate vaccines[30–32].

To date, there are few reports of ZIKV infection of pregnant non-human primates (NHP). Fetuses of 4/5 pigtail macaques whose mothers were delivered repeated subcutaneous high-dose inoculation in second or third trimester showed ZIKV-associated brain lesions, similar to those seen in human fetuses[8], including ependymal injury in the posterior lateral ventricles and periventricular gliosis[25,33]. In four pregnant rhesus macaques, subcutaneously ZIKV inoculated at gestational periods equivalent to mid-first to early third trimester, maternal–fetal transmission as evidenced by ZIKV RNA and pathology in some tissues, but not fetal CNS, was observed[23].

In this exploratory study, we artificially bypassed maternal–fetal transmission by direct intra amniotic (IA) inoculation, similar to the established NHP models of other fetal infections (e.g., cytomegalovirus; simian immunodeficiency virus[34,35]), and a recent intrauterine mouse model of ZIKV that resulted in viral antigen in the fetal brain[16]. We previously demonstrated that lipopolysaccharide administered IA induced neuropathology in preterm fetal rhesus macaques, similar to brain injury reported in preterm infants exposed to chorioamnionitis[36]. Furthermore, ZIKV persists in the amniotic fluid of humans for prolonged periods[37]. In addition to IA infection of the fetus, we also intravenously (IV) inoculated the mother. The purpose of combined IA and IV inoculations was to ensure fetal infection at defined times using a minimal number of animals, since at the time these studies were initiated in early 2016, the frequency and timing of spontaneous maternal–fetal transmission in ZIKV-infected pregnant humans or rhesus macaques was unknown. We used the same 2015 Brazilian ZIKV isolate we previously employed to characterize viral kinetics and tissue distribution during acute infection of non-pregnant macaques[20]. Four pregnant females ranging from gestational day (GD) 41 to GD90 at the time of inoculation were selected because the gestation of rhesus macaques is ~165 ± 10 days, and IA inoculation is feasible starting at GD40. Three GD-matched control mothers were sham-inoculated and subjected to the same anesthesia and sampling schedule as the ZIKV-inoculated mothers to ensure that adverse outcomes did not result from experimental manipulations.

Here we show that ZIKV infection resulted in fetal death in the earliest inoculated animal, GD41. Fetuses inoculated later in gestation in the late first and second trimesters survived to near term, but all exhibited CNS lesions including calcification and loss of neural progenitor cells that paralleled ZIKV RNA detection in CNS tissues. Because the combined maternal and fetal ZIKV inoculation approach reproduced fetal CNS lesions observed in humans, further development and optimization of

this NHP model of ZIKV infection in pregnancy is relevant to further study the determinants of fetal ZIKV neuropathogenesis and to explore therapeutic intervention strategies.

## Results

**Experimental design and pregnancy outcomes.** Four pregnant macaques were each IV and IA inoculated once in the first or second trimester with 5 $\log_{10}$ PFU per route with a 2015 Brazilian strain of ZIKV, followed by frequent sample collection and monitoring (Fig. 1). Three pregnant GD-matched control mothers were sham-inoculated and sampled on the same schedule. Although we intended to monitor pregnancies to GD155 and then terminate for detailed tissue collection, the GD41-inoculated fetus (GD41 fetus) was observed to be dead on GD48, 7 days post inoculation (dpi). The mother was therefore euthanized 7 dpi for detailed analysis of fetal and maternal tissues. The pregnant animal inoculated at GD64 (GD64 mother) had intermittent placental bleeding detected by ultrasound and bloody amniotic fluid with bloody vaginal discharge starting 39 dpi, and spontaneously vaginally delivered a small but viable neonate at GD151, at which time both were euthanized. The pregnant GD50 and GD90 mothers and three control animals were all monitored to the planned experimental end point of GD155 when fetectomy was performed, followed by necropsy. The fetal head sizes for the GD50 and GD90 fetuses were within the California National Primate Research Center (CNPRC) colony average during pregnancy, with normal head sizes at birth. The weights of the whole fetus or neonate, placental, and brain tissues (Supplementary Fig. 1) were similar to GD-matched control fetuses. The biparietal diameter of the GD64 fetus was consistently 2 s.d. below the colony mean, but at birth, the head of this small infant (with body weight of 280 gram (g), compared to the 477–590 g weight range of the three control fetuses, which is within the macaque average of 488 for females and 506 g for males[38]), was proportionate in size to the body (skull-to-shoulder length ratio of 2.5 for ZIKV fetus compared to 2.4–2.9 for three control fetuses), suggesting overall stunted body growth but not microcephaly (Fig. 2b, d and Supplementary Fig. 1).

**Maternal infection produces extended viremias and sero-conversion.** None of the four pregnant animals showed clinical signs of ZIKV-related illness, including fever, lethargy, or weight loss compared to control mothers. Viremias in ZIKV-infected mothers ranged from 5 to 43 dpi (Figs. 2a and 3b). The vaginal bloody discharge in the GD64 mother likely contaminated urine collected from cage pans thereafter; by contrast, no ZIKV RNA was detected in urine aspirated from the urine bladder by cystocentesis or from the GD50 or GD90 mothers after 13 dpi (Supplementary Fig. 2). Excluding the GD41 mother who had no detectable ZIKV-neutralizing antibody at euthanasia 7 dpi, the other three pregnant animals developed ZIKV-neutralizing antibody responses beginning 5 dpi, IgM from 7 dpi that waned by 5 weeks post inoculation (pi), and IgG responses from 14 dpi that were still high at the end of the experiment, up to 15 weeks pi (Supplementary Fig. 3). The GD90 mother that had the longest duration of viremia (43 dpi) developed the lowest neutralizing antibody titers (Sup Fig. 3). Except for effects on placentation and fetal development (see below), the female macaques had no significant histopathologic findings in tissues collected at necropsy (Table S1). Even in the absence of detectable plasma viremias at euthanasia, all mothers had detectable ZIKV RNA in tissues, with a clear tropism for lymphoid tissues (Fig. 4 and Supplementary Fig. 5), similar to our previous observations in ZIKV-inoculated non-pregnant female macaques[20].

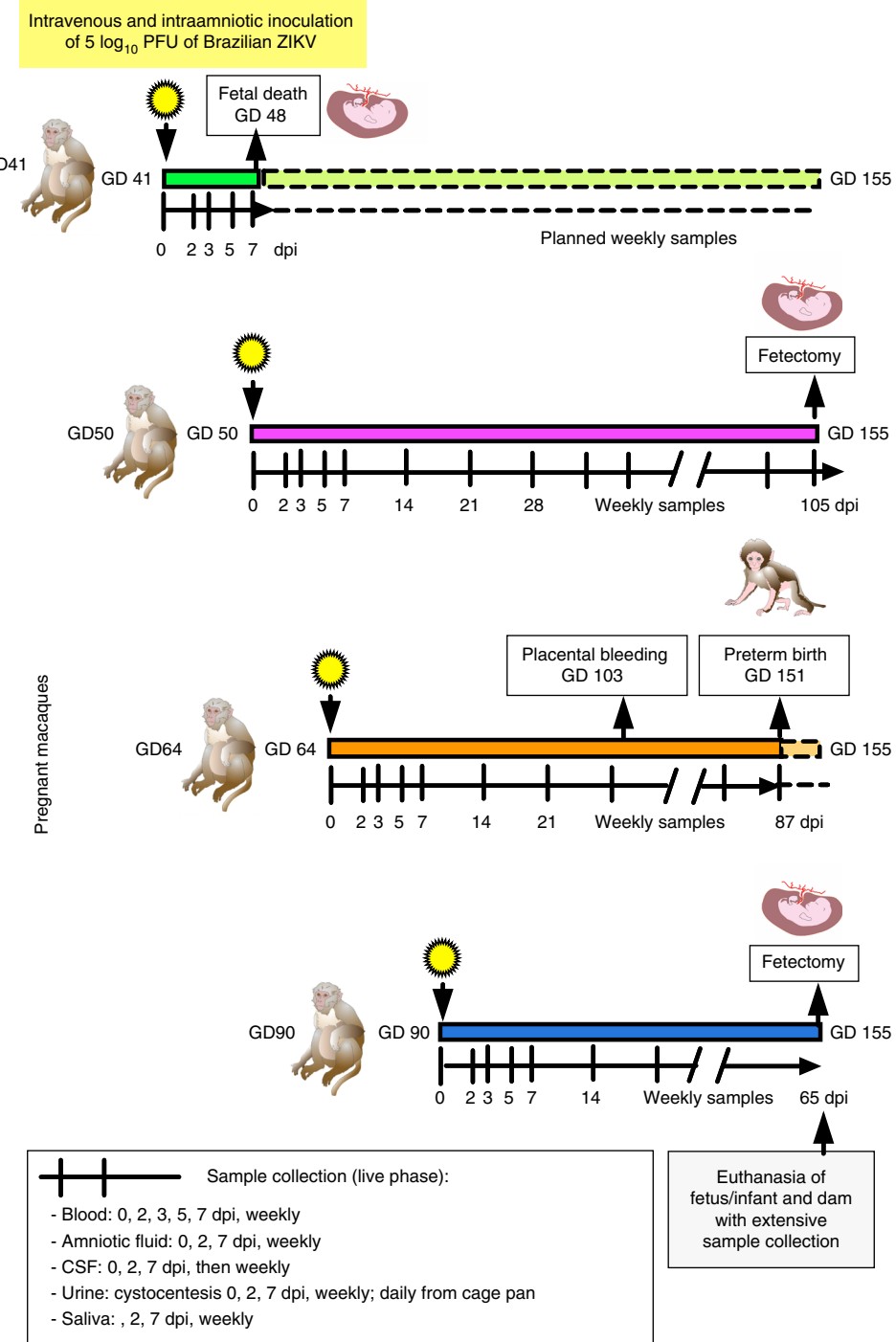

**Fig. 1** Schematic experimental design showing infection and sampling timeline for the four pregnant rhesus macaques in the study. Pregnant macaques, identified by gestation day (GD) of inoculation, were intravenously (IV) and intraamniotically (IA) inoculated with 5 $\log_{10}$ PFU of Brazilian ZIKV strain SPH/2015 in the first or second trimester of pregnancy. Sample collection and ultrasound-monitoring schedules, originally intended until near term at GD155, are noted. *CSF* is cerebrospinal fluid, dpi is days post inoculation. Animal color scheme is consistent for other figures. Not shown are the three GD-matched animals that were sham-inoculated and sampled on the same schedule as the ZIKV-inoculated macaques

**ZIKV infection leads to early death of GD41 fetus**. Although fetal abnormalities were not detected by ultrasound 2, 3, and 5 dpi, the fetal heartbeat of the GD41 animal was absent 7 dpi (Fig. 3a). The fetal crown-to-rump length was ~2.5 cm, where the average GD48 rhesus macaque fetal length by ultrasound is 3.2 cm[39]. The fetus showed moderate autolysis, likely due to in utero death within the previous 2-day period. The mother showed no gross lesions, but microscopic analysis revealed mild placentitis where the primary and secondary discs of the placenta had limited multifocal necrosis in the trophoblastic layer of the fetal chorionic plate that were replaced by fibrin and neutrophils (Table S1). Amniotic fluid collected at necropsy showed no anaerobic or aerobic bacterial growth, including *Listeria*, excluding ascending or iatrogenic bacterial infection as a possible cause of fetal death. This mother had the shortest ZIKV viremia duration in plasma compared to the other three mothers (Figs. 2a

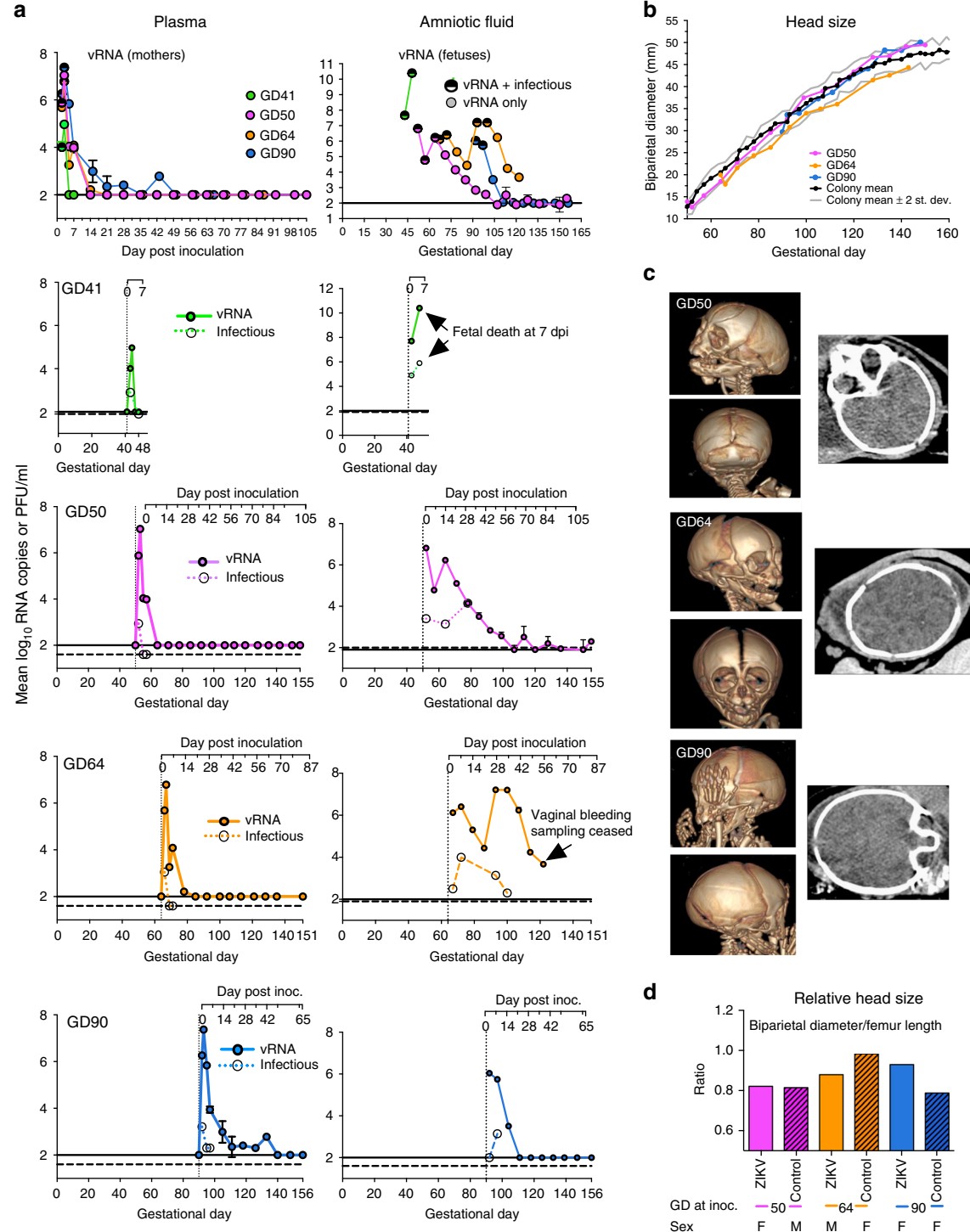

**Fig. 2** Pregnant macaques inoculated IV and IA develop prolonged ZIKV viremias and ZIKV RNA in amniotic fluid but show no fetal head growth restriction. **a** Viremia and ZIKV RNA levels in plasma and amniotic fluid of ZIKV-infected pregnant rhesus macaques: Each line shows ZIKV RNA (solid lines) or infectious virus (dotted lines) kinetics for a single animal, detected in mean $log_{10}$ RNA copies assayed in triplicate with error bars showing standard deviations or mean $log_{10}$ Vero cell plaque forming units (PFU) per ml assayed in duplicate. The horizontal solid line shows the ZIKV RNA limit of detection (LOD) of 2.0 $log_{10}$ RNA copies and the horizontal dotted line shows the plaque assay LOD of 1.6 or 1.9 $log_{10}$ PFU per ml. For the GD64 mother, due to recurring placental bleeding, amniotic fluid was no longer collected after 56 dpi (GD 120). **b–d** Most ZIKV-infected fetuses show normal growth as measured by head size determined by **b** the biparietal diameter at weekly ultrasound compared to the CNPRC colony mean (black line) ±2 s.d. (upper and lower gray lines), which was based on 10–100 fetuses reported earlier[60], **c** computed tomography (CT) scans at GD 141–148 and **d** relative head size calculated as the biparietal diameter divided by the femur length

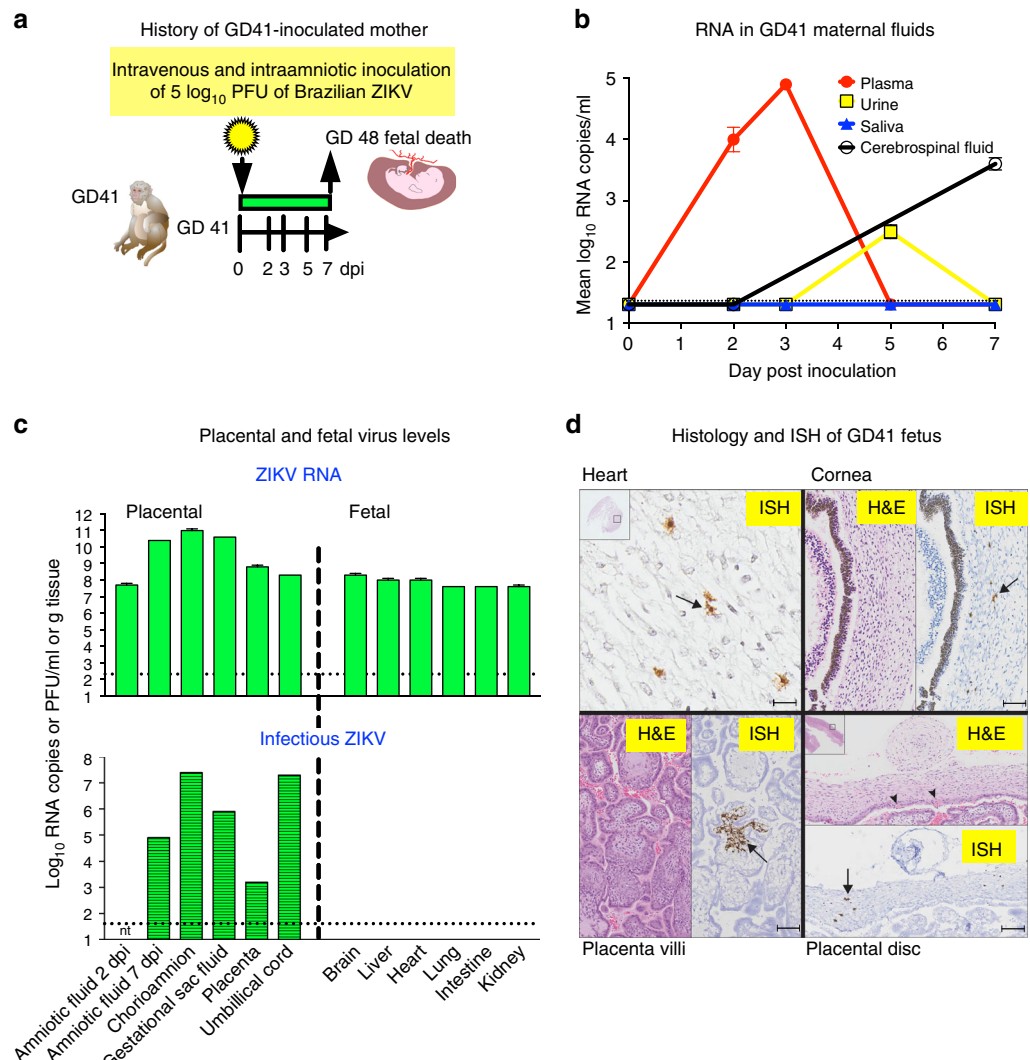

**Fig. 3** ZIKV infection associates with early death of GD41 fetus. **a** Schematic representation of infection outcome that resulted in fetal death 7 dpi. **b** ZIKV RNA levels in maternal fluids sampled by venipuncture, cystocentesis, saliva eluted from cotton swabs placed in the cheeks of macaques, or cervical spinal tap, reported in mean $\log_{10}$ RNA copies/ml and assayed in triplicate with error bars showing standard deviations. The dotted line shows the LOD, 2.3 $\log_{10}$ RNA copies/ml. **c** ZIKV RNA and infectious virus in placental and fetal tissues 7 dpi (except where noted for amniotic fluid), detected in mean $\log_{10}$ RNA copies assayed in triplicate or $\log_{10}$ Vero cell plaque forming units (PFU) per ml or gram tissue. The dotted lines show the ZIKV RNA LOD of 1.3 $\log_{10}$ RNA copies or the plaque assay LOD of 1.6 $\log_{10}$ PFU per ml or gram tissue. Amniotic fluid from 2 dpi was not tested (nt) by plaque assay. Error bars on RNA measures show standard deviations for three replicates. **d** Hematoxylin and eosin and in situ hybridization images of selected fetal tissues showing ZIKV RNA labeled brown and highlighted with arrows. Scale bars are as follows: heart, 54 μm, cornea, 150 μm, placenta villi, 130 μm, placental disc, 200 μm

and 3b), but experienced a similar magnitude and tropism of ZIKV RNA in tissues (Supplementary Fig. 4; fewer maternal tissues were collected since early fetal death was unexpected) compared to the other mothers (Supplementary Fig. 5). By contrast, ZIKV RNA levels in the amniotic fluid of the GD41 fetus increased from 7.7 $\log_{10}$ at 2 dpi to 10.4 $\log_{10}$ RNA copies/ml at 7 dpi (Fig. 2a); levels on both days were higher than in any of the other animals. ZIKV RNA was also detected in all 12 placental and fetal tissues and fluids sampled, with titers ranged from 7.6 to 11 $\log_{10}$ RNA copies/ml or gram of tissue (Fig. 3c), higher than the highest RNA level (6.8 $\log_{10}$ RNA copies/g) observed in its mother (Supplementary Fig. 4) or in fluids or tissues from any of the other mothers or fetuses (Fig. 4 and Supplementary Fig. 5). All placental tissues from the GD41 fetus also contained infectious ZIKV, ranging from 4.3 (in placenta) to 7.4 (in chorioamnion) $\log_{10}$ PFU/g tissue (Fig. 3c). By contrast, no infectious ZIKV was isolated in the fetal brain, liver, heart, lung, intestine, or

kidney (Fig. 3c), despite high levels of viral RNA in all tissues, possibly because of severe autolysis due to up to 2 days of fetal decay that may have destroyed ZIKV infectivity. In situ hybridization revealed ZIKV RNA in heart, cornea, placenta villus, placental discs (Fig. 3d), the fetal head and body, as well as lungs and kidney (Fig. 3d and Supplementary Fig. 6). No inflammatory response detected by histologic assessment in fetal tissues was noted, possibly reflecting limited immunological development at that early gestational stage or the acute time of infection, which could have preceded an immune response[40]. The absence of common confounding abortogenic bacterial pathogens, the increase in ZIKV RNA in amniotic fluid after IA inoculation, isolation of infectious ZIKV from placental tissues, diffuse ZIKV RNA detected by qRT-PCR exceeding inoculum levels in some tissues, and ISH showing ZIKV RNA in parenchymal cells in multiple organs in many fetal tissues support fulminant ZIKV replication that caused the death of the GD41 fetus.

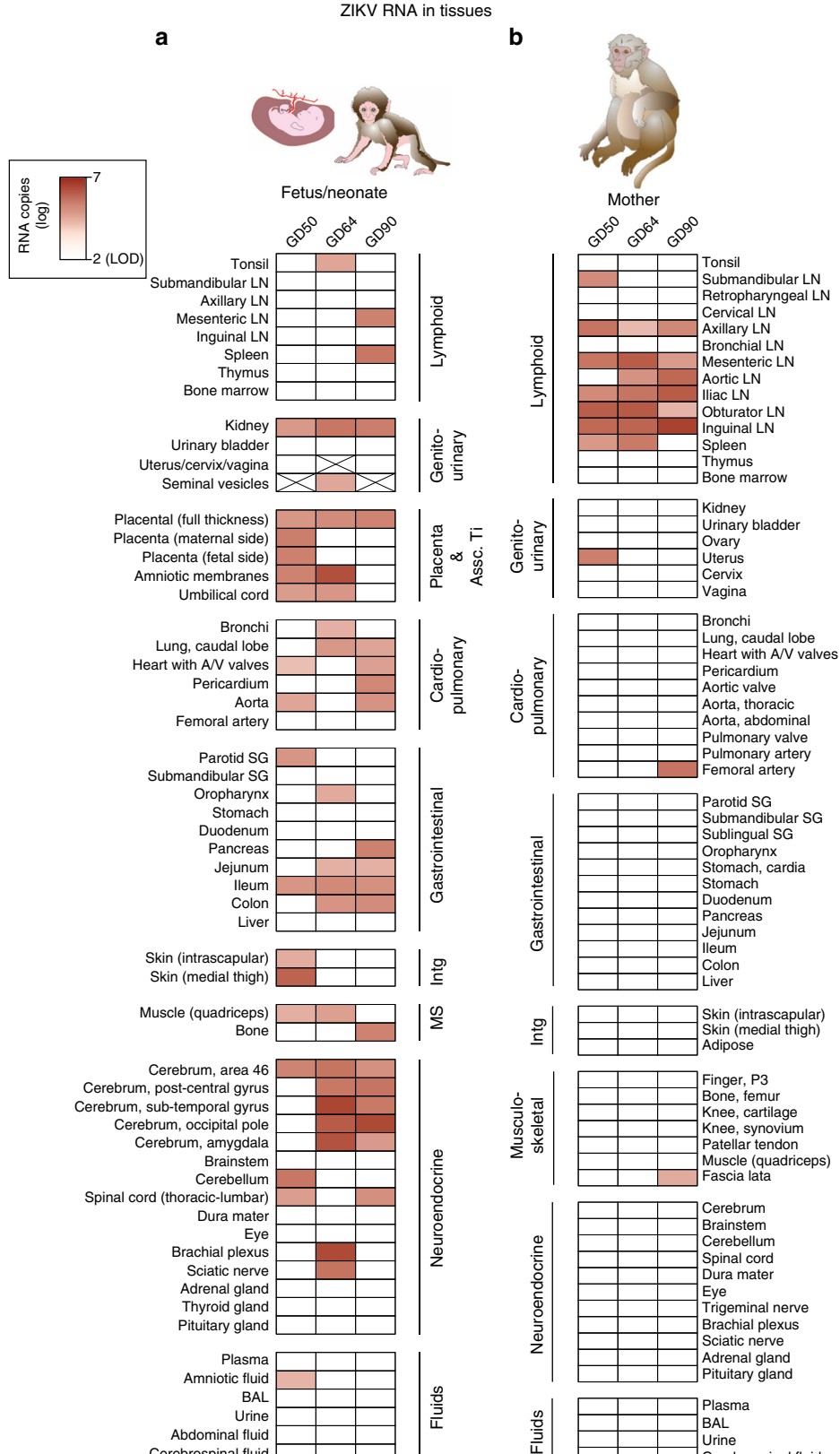

**Fig. 4** Tissue tropism of ZIKV RNA differs in fetal macaques compared to their mothers. ZIKV RNA in **a** fetal/neonatal and **b** maternal tissues categorized by system. The intensity of red highlights the quantity of ZIKV RNA detected. Absolute ZIKV RNA levels in individual tissues for these animals are shown in Supplementary Figures 5 and 7. Fetal/infant tissues that were not available due to their sex (GD50 and GD90 fetuses were females, GD64 infant was a male) are crossed out. LN lymph node, SG salivary gland, MS musculoskeletal, intg integument, BAL bronchioalveolar lavage, A/V atrioventricular, Assc Ti associated tissues

**Placental ZIKV infection at the end of gestation**. Placentas from the three animals inoculated at GD50, 64, or 90 had high levels of ZIKV RNA (Fig. 4 and Supplementary Fig. 7). Furthermore, placentas from all four fetuses/neonate in this study contained infectious ZIKV, with titers ranging from 1.6 to 3.9 $\log_{10}$ PFU/g (Supplementary Fig. 9). While the GD50 and GD90 placentas showed no histologic abnormalities, the GD64 fetus had evidence of suppurative placentitis, pneumonia, and dermatitis, secondary to possible bacterial infection and possibly placental abruption. Since the GD64 infant was spontaneously birthed, the placenta could not be collected in a sterile manner. These factors could have contributed to premature birth of the GD64 neonate. In situ hybridization showed ZIKV RNA in the septal stroma of scattered villi in a network of reticular cells, presumed placental macrophages and spindle cells within the chorionic plate (Fig. 3d, GD41 animal; other animals similar). To determine whether

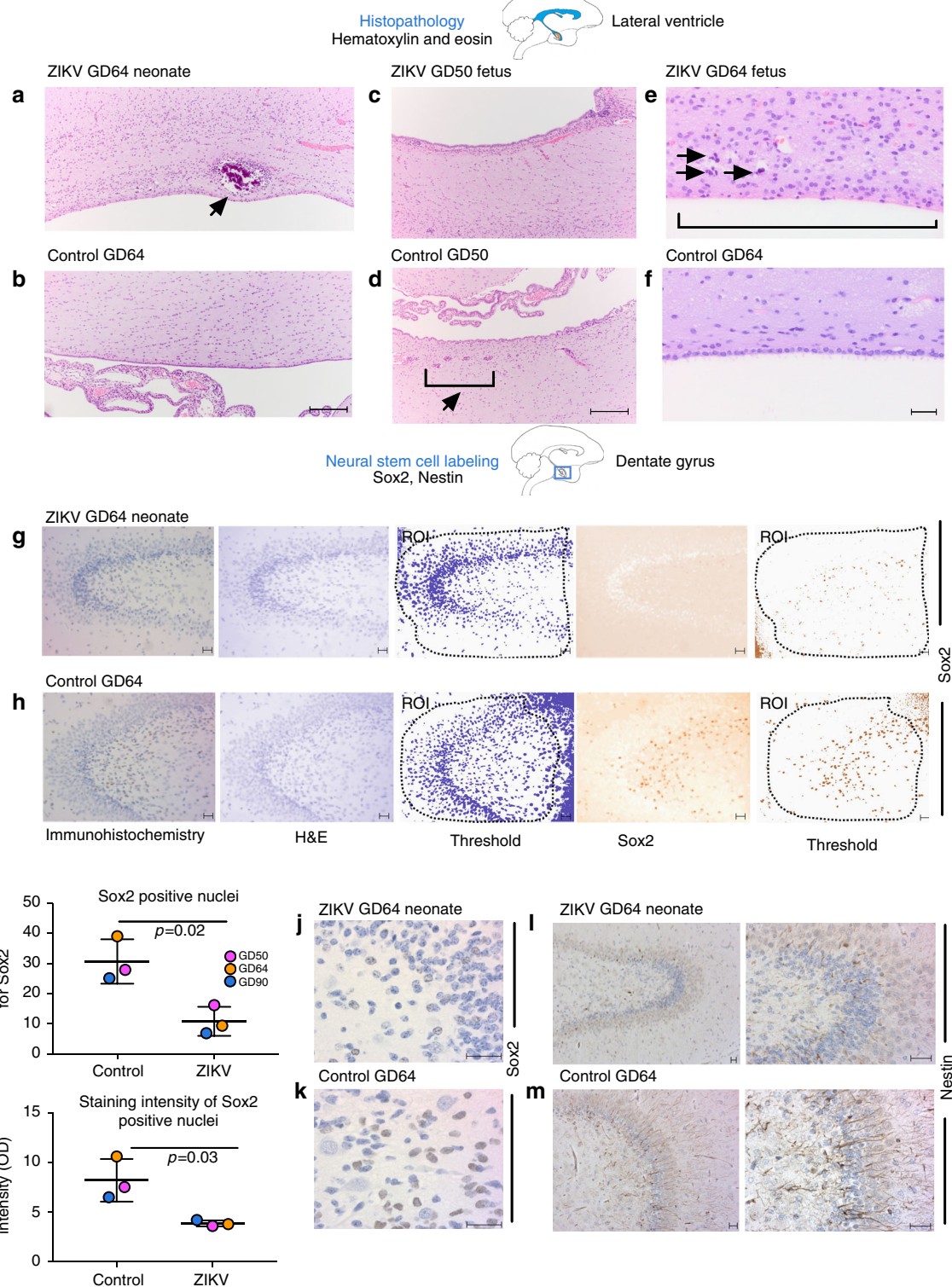

**Fig. 5** ZIKV infection results in fetal brain pathology. **a–f** Hematoxylin and eosin staining of brain tissues in the lateral ventricle from GD50 fetus and GD64 neonate with GD-matched control sections showing **a** calcification with gliosis (arrow) in ZIKV brain but absent in **b** control brain, **c** loss of rests of neural precursors in ZIKV brain contrasted with **d** normal rests of neural precursors (parenthesis/arrow) in control, scale bars show 100 μm and **e** complete absence of ependymal cells (parenthesis) with mineralization (arrows) in ZIKV brain compared to **f** control that shows intact lining and no mineralization, scale bars show 20 μm; **g, h, j–m** Immunohistochemistry for neural stem cells staining nuclei with sox2 (brown color) in **g** ZIKV GD64 brain compared to **h** GD64 control brain; Images are deconvoluted to independently show hematoxylin and sox2 staining, and thresholds for determining positive labeling are shown with regions of interest (ROI) highlighted, showing reduced labeling of neural stem cells in ZIKV infection in the dentate gyrus, scale bars show 20 μm; and **i** quantification of significantly reduced (Graphpad Prism, unpaired t-tests) Sox2-positive nuclei (upper) and intensity of Sox2 staining (lower) in three ZIKV brains compared to GD-matched controls, long horizontal line shows mean and shorter lines show standard deviation; high magnification Sox2 labeling of **j** ZIKV GD64 brain compared to **k** GD64 control brain; and nestin (brown) staining of **l** ZIKV GD64 brain compared to **m** GD64 control brain, cortex; scale bars for **j–m** are all 20 μm. Brain sketches show regions imaged

ZIKV infection modified immune responses in placental tissues, proinflammatory cytokine levels and leukocyte frequencies were characterized. Mean proinflammatory cytokine mRNA levels of IL-1ß, IL-6, IL-8, and CCL2 were slightly higher in the extra-placental fetal membranes (chorioamnion decidua tissue) from ZIKV animals compared to controls but no difference in TNF-α mRNA levels was observed (Supplementary Fig. 10). The relative proportion of leukocytes in the chorio-decidua of ZIKV animals vs. controls was not markedly different (Supplementary Fig. 11).

**Fetal ZIKV neurotropism and neuropathology at the end of gestation.** Consistent with detection of ZIKV RNA and infectious virus in placentas at the end of gestation, the three fetuses that survived to near term had detectable ZIKV RNA in amniotic fluid for at least 28 dpi, indicating ongoing virus replication (Fig. 2a). Amniotic fluid ZIKV RNA levels decreased in the third trimester, possibly due to transplacental passive transfer of maternal ZIKV IgG into the fetal circulation, supported by IgG but not IgM in plasma from the GD64 neonate at birth (Supplementary Fig. 3d). The GD64 amniotic fluid also had detectable ZIKV IgG at 57 dpi, possibly reflecting contamination from maternal blood given that leaking amniotic fluid was noted, suggesting a maternal–fetal breach. All three fetuses/neonates had detectable ZIKV IgG but no IgM and no neutralizing (from plaque reduction neutralization tests (PRNT)) antibody in cerebrospinal fluid (CSF) at necropsy. By contrast, neutralizing antibody was present in fetal and neonatal plasma (Supplementary Fig. 3d). The absence of IgM or neutralizing antibody in fetal/neonatal CSF may be a reflection of the timing of sampling in relationship to the developing blood–brain barrier and local antibody synthesis.

Fetal/neonatal tissues at necropsy 65, 87, or 105 dpi (for GD90, GD64, and GD50 inoculated animals, respectively) revealed a diffuse ZIKV RNA tropism that targeted neural, cardiopulmonary, urogenital, and lymphoid systems, and that contrasted with the lymphoid tropism observed in mothers (Fig. 4 and Supplementary Figs. 5 and 7 for ZIKV RNA levels, Supplementary Fig. 8 for ZIKV RNA in situ). Multiple central nervous system tissues from all term fetal/neonate brains contained detectable ZIKV RNA levels with several cerebral regions of the GD64 neonate and GD90 fetus exceeding 5 $\log_{10}$ RNA copies/g tissue (Supplementary Fig. 7). Infectious ZIKV was not detected in any of the fetal/neonatal brain sections or selected other tissues that contained more than 5 $\log_{10}$ RNA copies/g tissue, except for placentas.

The two near-term fetuses inoculated at GD50 and GD90, and the GD64 neonate all exhibited histologic changes in the brain. The periventricular parenchyma of the GD64 neonate showed areas of calcification surrounded by cuffs of gliosis not present in the control brain (Fig. 5a, b). Additionally, all fetal/neonatal brains had periventricular and subventricular loss of normal clusters of neural precursors present in the control brain (Fig. 5c, d, representative GD50 section shown). All three also had segmental loss of ependymal lining cells of the lateral ventricles in

all cerebral sections examined with periventricular bands of gliosis (Fig. 5e, f, representative GD64 fetal section shown). In situ hybridization (ISH) revealed very rare ZIKV-infected cells in some brain sections that were also ZIKV RNA positive by qRT-PCR (Supplementary Fig. 12). Other brain sections with detectable ZIKV RNA lacked an ISH signal. Immunohistochemical staining (Fig. 5g–k) of the dentate gyrus revealed significantly reduced numbers of immature neural stem cells in brains from ZIKV-infected animals, as visualized by expression of Sox2 (a nuclear stem cell transcription factor). In addition to a reduced number of sox2-positive cells, there was also a significantly decreased mean intensity of Sox2 expression in neural stem cells in ZIKV-infected compared to control animals. This observation was restricted to the dentate gyrus of the hippocampus and was not observed in subventricular zone of ZIKV vs. control brains with Sox2 staining (Supplementary Fig. 13). Neural stem cell numbers expressing nestin (an intermediate filament of neural stem cells) in the ZIKV GD64 dentate gyrus compared to the control brain (Fig. 5l, m) and in the cortex labeled by Tbr2 and Otx1 (Supplementary Fig. 14) were also qualitatively lower. Together, these data demonstrate that IV and IA inoculation of pregnant rhesus macaques leads to ZIKV infection of the fetal/neonatal CNS accompanied by neuropathologic changes including reduced neural stem cells in selected areas of the brain.

## Discussion

This study underscores the utility and relevance of our rhesus macaque model of ZIKV infection, by extending its value to investigate fetal infection and neuropathogenesis. We demonstrated that combined IV and IA inoculation of pregnant rhesus macaques during the first or second trimester with a 2015 Brazilian strain of ZIKV produced maternal and fetal infection in the absence of maternal disease, but early fetal death or neuropathologic changes in near-term fetuses reflecting some of the manifestations in congenital ZIKV syndrome.

Fetal death in the first trimester was accompanied by high ZIKV RNA levels in amniotic fluid and fetal tissues and high infectious virus levels in all placental tissues compared to the IA inoculum. Although only the GD41 fetus died in this study, ZIKV as a cause of fetal death is supported by observations of elevated rates of early fetal losses (5 out of 18 fetuses) in our ongoing studies with first-trimester pregnant macaques inoculated subcutaneously or via combined IV and IA routes with a 100-fold lower inoculation dose than the one used in the current study. The fetal loss observed in 27% of the pregnant animals in all our cohorts may parallel rates of miscarriage in women, typically difficult to measure but currently being estimated in large cohort studies in ZIKV endemic areas. Furthermore, the growth restricted GD64 fetus whose mother experienced placental bleedings throughout the third-trimester parallels in utero growth restriction and placental abnormalities described in ZIKV-infected pregnant women[3].

Compared to the mothers where ZIKV RNA was detected in most lymphoid tissues, ZIKV RNA in fetuses and the neonate was predominantly neurotropic. The fetal neuropathologic changes observed here included ependymal lining cell loss, calcifications, and reduced neural progenitor cells in the same brain regions where ZIKV RNA was detected by qRT-PCR. These findings parallel observed calcifications in the periventricular white matter in human fetal brains[4,7,8,41–44]. The discordance between qRT-PCR and ISH signal in CNS tissues, where the region tested by each method was not necessarily adjacent, indicates that ZIKV RNA may be multifocally distributed or that ISH is less sensitive than qRT-PCR; a correlation between ISH intensity and RNA measured by qRT-PCR has not been established for ZIKV. The relatively low levels of ISH-reactive cells observed in ZIKV RNA-positive fetal macaque brains contrasts with intense ISH staining seen in some human fetal brains[8], possibly reflecting differences in timing of sampling or other virus–host interactions, which merit further study. The decrease in neural progenitor cells detected with Sox2 staining observed here and in human fetuses (reviewed in ref. [45]) could reflect earlier cytolytic ZIKV infection leading to reduced cell migration, restricted neurogenesis, and cell death, resulting in aberrant neurologic development.

The three near-term fetuses/neonate had evidence of ZIKV persistence in the CNS at a time when fetal blood, urine, and CSF, the most accessible samples in human infants, showed no detectable ZIKV RNA, highlighting challenges with diagnosing congenital ZIKV infection in human newborns. Furthermore, although IgG and neutralizing antibody were detected in CSF of all three offspring, these antibodies may reflect passive transfer from the mother across the placenta and then across a leaky blood–brain barrier or contamination from maternal or fetal blood. Unlike what has been reported in some human newborns[46], no ZIKV-specific IgM was detected in fetal CSF, but all three fetuses/neonates had ZIKV-specific IgG in CSF.

Although the fetuses in this study did not develop microcephaly within the time period of observation (i.e., end of gestation), microcephaly at birth is only observed in a small fraction of ZIKV-exposed human fetuses[8]. Our observations are consistent with the spectrum of outcomes observed in congenital Zika virus syndrome, including neurologic disease without microcephaly and death[8]. Although no eye pathologies were noted, the cornea of the GD41 fetus contained ZIKV RNA detected by ISH, suggesting virus infection of the eye, another manifestation in congenital Zika virus syndrome. The presence of ZIKV RNA and neuropathologic changes in fetal brains from all study subjects supports the idea that there may be major neurobiological consequences of fetal ZIKV infection in human neonates, infants, and children that do not manifest microcephaly at birth, consistent with recent observations in which neurologic disease caused by ZIKV infection arose post-partum[47]. Ongoing studies with macaque infants born from ZIKV-infected mothers will address this possibility. In the present study, detection of infectious ZIKV in the macaque placenta but not other selected maternal or fetal tissues up to 105 dpi supports the idea of a placental reservoir, and is similar to human data[2,48–55]. Further research is needed to determine whether testing placental tissue collected at time of delivery for the presence of ZIKV RNA, recently applied in the human clinic[56], could be a useful surrogate marker to predict infection of the neonate. Testing of placental tissues might be especially useful in cases where maternal serologic assays cannot distinguish ZIKV from related flaviviruses or when other neonatal test results from blood, urine, or CSF are inconclusive.

Although the direct fetal IA inoculation route artificially bypasses natural transplacental ZIKV transmission, it also circumvents the low ≈6% human frequency for congenital ZIKV

malformations estimated from US women[57] and the variable timing of mother-to-child transmission, thereby reducing the number of study subjects required. This IA model more closely parallels human fetal ZIKV-induced neuropathogenesis[8] than a related study that used subcutaneous maternal inoculation of four rhesus macaques, since in that model, no fetal CNS infection or neuropathologic changes were observed[23]. Our data mirror observations from pigtail macaques inoculated subcutaneously with artificially high ZIKV doses (five inoculations each at 7 $\log_{10}$ PFU, where ZIKV-infected mosquitoes typically transmit 1–5 $\log_{10}$ PFU[58,59]), whose fetuses also developed ependymal cell injury as well as other severe neurologic outcomes including cerebral hypoplasia, periventricular white matter gliosis, axonal cell injury, and reduced neural stem cells[25,33]. Together, these studies show that the IA route or artificially high ZIKV dosing may be required to consistently produce fetal neurologic infection and disease in macaques.

In conclusion, by demonstrating that IA ZIKV inoculation provides a method to consistently induce fetal neuropathogenesis, the current study is a step toward development of a relevant animal model of fetal ZIKV neurologic disease. The direct fetal inoculation route can be used to study viral and host determinants of fetal pathogenesis including the role of timing of infection, viral or host genetics, prior or concomitant co-infections, and to test the efficacy of antiviral drugs or monoclonal antibodies selected to cross the placenta to inhibit virus replication in the fetus. Thus, together with developing macaque models that mimic natural transplacental transmission[23,25], further development and optimization of the direct fetal inoculation model presented here can address remaining key questions in ZIKV pathogenesis and provide important proof-of-concept information on the efficacy of interventions to guide clinical trials.

## Methods

**Animals and care.** All of the macaques in the study were from the conventional breeding colony and born at the California National Primate Research Center (CNPRC), and were type D retrovirus-free, SIV-free and simian lymphocyte tropic virus type 1 antibody negative. In addition, they were confirmed to be antibody negative for West Nile virus (WNV) using the simian WNV ELISA (Xpress Bio). Animals were not screened for dengue virus antibody given the absence of local circulation in California. An additional three macaques were gestation day (GD) matched control placebo-inoculated animals. All animals had previously successful pregnancies with live births (range, 1–6). Animals were selected from the CNPRC's timed breeding colony. Gestational ages were determined from the menstrual cycle of the dam and the fetus length at initial ultrasound compared to growth data in the CNPRC rhesus macaque colony[60]. Pregnant animals were housed and all experimental procedures were performed at the CNPRC that is accredited by the Association for Assessment and Accreditation of Laboratory Animal Care International (AAALAC). Animal care was performed in compliance with the 2011 *Guide for the Care and Use of Laboratory Animals* provided by the Institute for Laboratory Animal Research. Macaques were housed indoor in stainless steel cages (Lab Product, Inc.) and were exposed to a 12-h light/dark cycle, 65–75 °F, and 30–70% room humidity. Animals had free access to water and received commercial chow (high protein diet, Ralston Purina Co.) and fresh produce supplements. The study was approved by the Institutional Animal Care and Use Committee of the University of California, Davis.

**Sample collection and clinical observations and monitoring.** Macaques were evaluated twice daily for clinical signs of disease including poor appetence, stool quality, dehydration, diarrhea, and inactivity. When necessary, macaques were immobilized with ketamine hydrochloride (Parke-Davis) at 10 mg/kg and injected intramuscularly after overnight fasting. Animals in both the ZIKV-treated and placebo cohorts were sedated at time zero (time of virus inoculation), days 2, 3, 5, and 7, and then weekly for sample collection and ultrasound monitoring. Blood samples were collected using venipuncture. Cerebrospinal fluid was collected via a cervical spinal tap. Urine was collected on days of sedation via cystocentesis or from pans under cages. Amniocentesis was conducted via ultrasound guidance. Rectal temperature was measured at every sedation. Since frequent anesthesia and amniocentesis are risk factors for pregnancy loss, the three non-infected pregnant macaques underwent the same schedule of procedures as ZIKV-inoculated animals; all had normal fetal development. Ultrasound-guided amniocentesis was

conducted using sterile techniques by inserting a 22 gauge, 1.5-inch spinal needle into the amniotic sac at the gestation days (GD) noted in Fig. 1. The fetal heart rate was obtained before and after amniocentesis. The area of umbilical entry through the amniotic sac was always avoided to prevent damage to the umbilical arteries and vein. Whenever possible, placental tissue was avoided during the collection of amniotic fluid. All samples were archived immediately at −80 °C for viral RNA and infectious virus assays.

**Inoculations.** A 2015 isolate of ZIKV from Brazil (strain Zika virus/H.sapiens-tc/BRA/2015/Brazil_SPH2015; genbank accession number KU321639.1), the same strain we characterized in non-pregnant animals[20], was used. The strain was isolated from the plasma of a transfusion recipient and was passaged once on Vero cells (CCL-81) recently purchased from the American Type Culture Collection and then titrated by Vero cell plaque assay. Deep sequencing of the virus inoculum revealed it was mycoplasma-negative and not contaminated by nucleic acid sequences from other microorganisms. The consensus genome of the ZIKV inoculum was identical to the genbank sequence. Aliquots were stored in liquid nitrogen and each time thawed right before the inoculation procedure. The inoculum was adjusted to 5.0 $\log_{10}$ PFU (corresponding to 7.8 $\log_{10}$ RNA) in 1 ml of RPMI-1640 medium and injected intravenously (IV) to simulate direct IV mosquito feeding[61] in the saphenous vein of four adult pregnant animals under aseptic conditions. The same dose was also administered just after the IV inoculation via ultrasound-guided intra amniotic (IA) inoculation under aseptic conditions. Each mother was only inoculated on one day; inoculations occurred at GD 41, 50, 64, or 90 corresponding to first and second trimester of human gestation, where the normal gestation of rhesus macaques is 165 days. Three control placebo-inoculated pregnant animals had IV and IA injections of 1 ml of RPMI-1640 diluent at GD 50, 64, or 90 using aseptic conditions.

**Fetal measurements.** Fetal measurements were collected during pregnancy after mothers were sedated with ketamine hydrochloride (10 mg/kg) for sonographic assessments and amniocentesis. The biparietal diameter (BPD) was measured by ultrasound (US) performed by veterinary staff. Images were analyzed by a researcher blind to the ZIKV infection status of each animal. BPD was drawn in ImageJ and rescaled into metric units using the scale available on each image that was produced by the US machine. The CNPRC colony BPD mean was derived from historical data[60]. Computerized tomography (CT) scans were performed on a GE Medical Systems machine (Discovery 610), using a helical head sequenced acquisition protocol. After a localizer scan, a scan with the following parameters was deployed: 10.0 s 99.38 mm 0.62 mm 20.0 mm 1.0 1 X-ray source. CT X-ray source parameters: 1 120.0 kV 280.0 mA 280.0 mA 2 s; CT dose 100.79 mGycm IEC Head Dosimetry Phantom 1007.88 mGycm. Average skull volumes were computed in Osirix on all three imaging orientations (axial, sagittal, and coronal) and then averaged. Length of the clavicles was computed from the CT images and used to create a skull-to-shoulder ratio. At necropsy, head circumference (HC) was measured with a tape measure, fetal and organ weights with a scale, and crown-rump length, biparietal diameter, head height, head length, and femur length, were derived from caliper measurements.

**Necropsy and tissue collection and processing.** At necropsy, fetal and maternal tissues were surgically removed. All necropsies were performed by a board-certified pathologist and two technicians. Hysterotomy was performed on the pregnant macaques under inhalation anesthesia, and the fetus, placenta, fetal membranes, umbilical cord, and amniotic fluid were collected for detailed tissue dissection post-fetal euthanasia with an overdose of sodium pentobarbital (≥120 mg/kg). Shortly after, the mother was euthanized with an overdose of sodium pentobarbital (≥120 mg/kg). Each tissue was grossly evaluated in situ, and then excised, with further dissection with separate forceps and razor blades for each tissue to minimize risks for cross-contamination. Tissues were collected snap-frozen in Dulbecco's minimum essential medium (DMEM) and RNAlater and homogenized to a liquid state[20] with glass beads (Fisher Scientific) or a 5 mm steel ball (Qiagen). If tissue remained after snap freezing and RNAlater archival, tissues were preserved in 10% neutral-buffered formalin and routinely paraffin-embedded and processed with hematoxylin and eosin (H&E) or as noted below.

**Isolation and quantitation of viral RNA from fluids and tissues.** Zika virus RNA was isolated from samples and measured by qRT-PCR according to methods described previously[20] modified to increase the initial volume of sample tested from 140 to 200 µl (when available) to increase sensitivity. The limit of detection for plasma, amniotic fluid (AF), and urine of viral RNA copies varied from 1.6 to 2.0 according to the volume tested. For organs, the limit of detection varied depending on the weight of tissue sampled and volume of DMEM needed to homogenize to liquefaction, with a mean of 2.3 $\log_{10}$ RNA copies. For selected samples from dams, the Grifols Procleix® ZIKV assay was used on a higher volume (0.5 ml) of plasma or tissue homogenate. The Procleix ZIKV assay® is a qualitative in vitro assay that detects ZIKV RNA on the fully automated Panther® system. The assay is CE-marked (European Conformity) and is currently being used for screening-donated blood in the United States under an investigational new drug

protocol. All Procleix® assay steps were performed with an internal control and a limit of detection of 0.6 $\log_{10}$ copies/ml.

**Infectious virus quantification by plaque assay.** Infectious virus in fluids and tissues was titrated via tenfold dilution using a Vero cell plaque assay, as described earlier[20]. Plaque assays were not performed on samples that contained less than <3 $\log_{10}$ genomes since our previous work showed those samples would not show detectable plaques. The limit of detection was 1.6–1.9 $\log_{10}$ plaque forming units (PFU) per ml fluid or gram tissue.

**Neutralizing antibody quantification by plaque reduction neutralization test (PRNT).** End point 80% PRNT titers in macaque plasma were determined on serial twofold dilutions using a plaque reduction neutralization test (PRNT) described earlier[20]. The antibody titer is reported as the reciprocal of the highest dilution of plasma, indicated as the final virus-serum dilution that inhibited at least 80% of plaques.

**Qualitative Zika IgG and IgM antibody detection by ELISA.** Plasma samples from selected time points were tested at a 1:50 dilution for ZIKV antibody using a commercially available ELISA (Xpress Bio), according to the manufacturer's instructions. Since the conjugate supplied by the manufacturer is anti-IgG, the assay detects mainly ZIKV-specific IgG. The same ELISA plate was also used to measure ZIKV-specific IgM by replacing the conjugate with an anti-monkey IgM conjugate (Kirkegaard & Perry Laboratories, Inc.).

**Quantitative Zika IgG antibody titers by ELISA.** High-binding 96-well ELISA plates (Greiner) were coated with 30 ng 4G2 antibody (clone D1-4G2-4-15) in carbonate buffer, pH 9.6 overnight at 4 °C. Plates were blocked in Tris-buffered saline containing 0.05% Tween-20 and 5% normal goat serum for 1 h at 37 °C, followed by an incubation with Zika virus (strain PRVABC59, BEI) at $8.5 \times 10^4$ focus forming units/well for 1 h at 37 °C. Heat inactivated plasma was tested at a 1:12.5 starting dilution in eight serial fourfold dilutions, incubating for 1 h at 37 °C. A horseradish peroxidase (HRP)-conjugated goat anti-monkey IgG antibody (AbCam) was used at a 1:2500 dilution, followed by the addition of SureBlue reserve TMB substrate (KPL, Gaithersburg, MD). Reactions were stopped by stop solution (KPL, Gaithersburg, MD). Optical densities were detected at 450 nm. The $\log_{10}$ 50% effective dilutions (ED$_{50}$) were calculated for IgG-binding responses against ZIKV whole virion.

**Viral RNA detection by in situ hybridization.** For in situ hybridization (ISH), probes were hybridized to the positive sense ZIKV RNA genome. One set was designed to detect both African and Asian ZIKV strains (genbank accession numbers: AY632535.2 and KU321639.1) and consists of 80 paired probes spanning the region from 139 to 4697 in a ZIKV reference genome (NC_012532.1). A second probe set that hybridizes perfectly (0 mismatches) to the 2015 Brazil ZIKV strain used in this study consists of 70 paired probes covering genomic nucleotides 150–4168. Colorimetric ISH was performed manually on superfrost plus slides (Fisher Scientific) using the RNAscope kit (Advanced Cell Diagnostics) according to the manufacturer's instructions. Each 5 µm section of formalin-fixed, paraffin-embedded tissue was pretreated with heat and protease prior to probe hybridization for 2 h at 40 °C. An unrelated probe of similar length and complexity, as well as a probe designed to detect bacterial dapB were used as controls on all slides. A horseradish peroxidase–based signal amplification system was hybridized to the target probes followed by color development with 3, 3′ -diaminobenzidine (DAB; Advanced Cell Diagnostics). Slides were counterstained with hematoxylin and mounted with xylene-based SHUR/Mount (Triangle Biomedical Sciences). Pathologists interpreting slides were blinded to ZIKV-infected and control animal identifiers. Positive staining was identified as brown, punctate dots.

**Neural cell immunohistochemistry (IHC).** Slides containing fetal brain tissues were rehydrated and subjected to heat-induced antigen retrieval using target antigen retrieval solution (Dako). Primary antibodies were incubated overnight, followed by labeling with biotinylated secondary antibodies and horseradish peroxidase-conjugated neutravidin. Colorimetric detection was performed using 3,3′-DAB and slides were counterstained with hematoxylin prior to mounting. The antibodies used for IHC were: Sox2 from R&D MAB2018 at 10 µg/ml (mouse), Nestin: Covance pRb-315c (rabbit) at a 1:200 dilution, Tbr2: Biorbyt orb158576 (rabbit) at a 1:200 dilution, and Otx1/2: Abcam 21990 (rabbit) at 5 µg/ml. Pathologists interpreting slides were blinded to ZIKV-infected and control animal identifiers. For IHC quantation, original-labeled images were deconvoluted using ImageJ (NIH) to separate hematoxylin (blue; staining all nuclei) and DAB chromagen (brown; sox2-positive nuclei) conditions. Color thresholds were set to identify positive labeling above background and colocalization analyses were performed in ImageJ in user-defined regions of interest (ROI) using the coloc2 plugin and a Costes P-value of 1, indicating that colocalization is not random. The percentage of colocalized pixels for each condition and the mean DAB signal intensity (sox2) was measured. The percentage of nuclei-expressing sox2 and the mean pixel

intensity for Sox2-positive nuclei were compared for ZIKV and control images by two-tailed *t*-tests.

**Chorioamnion decidua cell suspensions**. Purified decidua cell suspensions were prepared as previously described[62]. Briefly, the membranes were dissected from each placenta. Decidua parietalis was scraped from the underlying chorion, washed and digested at 37 °C with 125 mg/100 ml Dispase-II (Life Technologies, Grand Island, NY) plus 50 mg/100 ml collagenase A (Roche, Indianapolis, IN) in DMEM: F12 medium with antibiotics. After 30 min, 0.4 mg/100 ml DNAse I (Roche) was added to the cell suspension for an additional 30 min at 37 °C on a shaking plat-form. Cell suspensions were filtered through 70 μm cell strainers twice, washed in PBS and counted. Red blood cell lysis was performed using ammonium chloride/potassium carbonate/ethylenediaminetetraacetic acid. The viability was >90% by trypan blue exclusion test.

**Flow cytometry on chorioamnion decidua cells**. For multi-parameter flow cytometry (LSR Fortessa, BD Biosciences, San Diego, CA), a cocktail of conjugated antibodies validated for the rhesus macaque was used (http://www.nhpreagents. org/NHP/clonelist.aspx?ID=15). For the majority of immunophenotyping studies, we used freshly isolated purified decidua cell suspensions. The following mAbs were used, all at a dilution of 1:20: anti-CD3 (SP34-2, anti-CD56 (NCAM16.2), anti-CD45 (D058-1283), (BD Biosciences); anti-HLA-DR (L243) (Biolegend, San Diego, CA); anti-CD14 (TUK14) (Life Technologies); anti-CD88 (P12/1) (AbD Serotec, Hercules, CA). Doublets were excluded on the basis of forward scatter properties and dead cells were excluded using LIVE/DEAD Fixable Aqua Dead Cell Stain (Life Technologies). Unstained and isotype control were used to determine positive staining for each marker. Data were analyzed using FlowJo software 9.5.2 (TreeStar Inc., Ashland, OR).

**Chorioamnion decidua cell RNA isolation and cDNA generation and quanti-tative RT-PCR**. Total RNA was extracted from snap-frozen chorioamnion decidua after homogenizing in TRIzol (Invitrogen). RNA concentration and quality were measured by a nanodrop spectrophotometer. Reverse transcription of the RNA was performed using Verso cDNA synthesis kit (Thermo-Scientific), following the manufacturer's protocol. Quantitative RT-PCR was carried out in a StepOnePlus Real Time PCR system (Life Technologies) following standard cycling conditions. Quantitative RT-PCRs were performed with rhesus-specific TaqMan gene expression primers (Life Technologies). The eukaryotic 18S rRNA (Life Technol-ogies) was endogenous control for normalization of the target RNAs and a sample from a placebo inoculated animal was used as a calibrator. The cytokine mRNA values for ZIKV-infected animals were expressed relative to the average value of the control group. Statistical analysis on cytokine mRNA values were done via *t*-test, with two-sided *p* values of <0.05 considered significant.

**Data availability**. The authors declare that the data supporting the findings of this study are available within the article and its Supplementary Information files, or are available from the authors on request.

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

## Acknowledgements

We acknowledge the research and support staff at CNPRC for their assistance with all aspects of the study, as well as the UC Davis IACUC and Environmental Health and Safety Offices for expedited approval for work on this project. Sarah Mills, Abigail Spinner, and the staff of the Pathogen Detection Laboratory, Clinical Laboratories and Colony Research Services at CNPRC helped with procedures, necropsies and biospecimen processing, and analysis. Kathy West provided illustrations. The Zika research team at the Wisconsin National Primate Research Center provided ongoing advice and data sharing. This study was supported by a CNPRC pilot research grant to L.L.C. and K.K.A. v.R., the Office of Research Infrastructure Programs/OD (P51OD011107), start up funds from the Pathology, Microbiology and Immunology Department, and 1R21AI129479-01 to K.K.A.v.R. This effort was partially funded by the United States Food and Drug Administration (FDA), via contract #HHSF223201610542P "Companion Studies to Define the Distribution and Duration of Zika virus Infection in Non-Human Primates". The views, opinions, and/or findings contained in this report are those of the authors and not necessarily those of the FDA.

## Author contributions

Conceptualization: L.L.C., K.K.A.v.R., E.B.M., R.I.K., C.C., K.C., J.M., and J.Y. Methodology: L.L.C., K.K.A.v.R., R.I.K., P.A.P., R.O., T.M., M.M., and R.M. Investigation: R.I.K., A.S., K.L.C., C.C., J.Y., J.W., J.U., J.M.L., K.G., A.A., K.W., L.L.C., K.K.A.v.R., E.B.-M., K.W., J.R.R., W.v.M., A.M.G., K.J., H.H., P.S., S.P., P.A.P., S.G.K., and V.A.H. Writing-original draft: L.L.C., K.K.A.v.R., and R.I.K. Writing-review and editing: L.L.C., K.K.A.v.R., R.I.K., P.P., K.W., S.G.K., R.O., and T.M. Visualization: L.L.C., K.K.A.v.R., A. A., R.I.K., P.A.P., K.W., and E.B.-M. Supervision and project administration: L.L.C., K.K.A.v.R. Funding acquisition: L.L.C. and K.K.A.v.R.

## Additional information

**Competing interests:** J.M.L. and K.G. are employees of Grifols Diagnostics Solutions, Inc. They tested selected maternal tissues for ZIKV RNA. They did not influence the interpretations in the manuscript. The remaining authors declare no competing interests.

