## [Peer Review File · Nature Communications]

Reviewers' comments:

Reviewer #2 (Remarks to the Author):

General

- The authors made considerable efforts to modify the manuscript based on previous reviews. These efforts strengthened the manuscript and clarified the data presented.

What are the major claims of the paper?

- The authors have restated their research claims with a focus on creating a pathogenesis versus transmission model and reducing their claims the model with broadly support evaluation of Zika countermeasures. The modifications regarding the claims are adequate.

Are they novel and will they be of interest to others in the community and the wider field?

- They model is exploratory and methodological challenges remain. That being said, it does add to the larger body of literature and will be interest to the wider field.

If the conclusions are not original, it would be helpful if you could provide relevant references.

- Referencing is adequate.

Is the work convincing, and if not, what further evidence would be required to strengthen the conclusions?

- Elements of CZS are evident in most animals but there is variation among the small sample size. The dual IV and IA inoculation method, despite the reasonable explanation for its use, weakens the overall model because it bypasses processes which may potentially contribute to overall pathogenesis.

On a more subjective note, do you feel that the paper will influence thinking in the field?

- Inform yes, influence no.

Please feel free to raise any further questions and concerns about the paper.

- The authors have done an adequate job addressing previous reviews to the extent the study methods allow.

We would also be grateful if you could comment on the appropriateness and validity of any statistical analysis, as well the ability of a researcher to reproduce the work, given the level of detail provided.

- The level of detail is appropriate.

Concern: a revised manuscript should provide quantification of Sox and nestin in fetal brain.

Response: Our revised manuscript includes quantification of neural stem cell labeling Sox2 in fetal brains. We added figure panels (Fig. 5i) showing that, consistent with the qualitative observations we noted in our first draft, all brains from ZIKV fetuses showed significantly fewer Sox2 labeled cells and significantly less intense Sox2 expression compared to the gestation day matched fetal control brains.

Attempts were made to quantify both the number of nestin positive cells and the intensity of nestin staining. However, the complex arrangement of nestin-positive cell processes (see Fig. 5l-m) made it impossible to accurately count positive cells. Additionally, the intensity measurements were not validated upon repeat measurement.

We therefore feel that quantification of Sox2-positive nuclei is reflective of the relative numbers of neural stem cells in control versus ZIKV brains.

The manuscript was modified (**in bold**) to reflect the added quantitative Sox2 data. Specifically:

Abstract (line 75): “The fetal and neonatal brains also exhibited calcifications and **reduced neural precursor cell levels**”.

Results (lines 272 and following): “Immunohistochemical staining (**Fig. 5g-k**) of the dentate gyrus revealed significantly reduced numbers of immature neural stem cells in brains from ZIKV-infected animals, as visualized by expression of Sox2 (a nuclear stem cell transcription factor). In addition to a reduced number of sox2-positive cells, there was also a significantly decreased mean intensity of Sox2-expression in neural stem cells in ZIKV-infected compared to control animals. This observation was restricted to the dentate gyrus of the hippocampus and was not observed in subventricular zone of ZIKV versus control brains with Sox2 staining (Sup. Fig. 13). Neural stem cell numbers expressing nestin (an intermediate filament of neural

stem cells) in the ZIKV GD64 dentate gyrus compared to the control brain (Fig. 5l-m) and in the cortex labeled by Tbr2 and Otx1 (Sup. Fig. 14) were also qualitatively lower. Together, these data demonstrate that IV and IA inoculation of pregnant rhesus macaques leads to ZIKV infection of the fetal/neonatal CNS accompanied by neuropathologic changes **including reduced neural stem cells in selected areas of the brain.**

Discussion (line 335 and following): “The decrease in neural progenitor cells **detected with Sox2 staining observed here** and in human fetuses (reviewed in reference 45) could reflect earlier cytolytic ZIKV infection leading to reduced cell migration, restricted neurogenesis, and cell death, resulting in aberrant neurologic development.”

We also added discussion of new data from a parallel study published recently (Adam-Waldorf et al., *Nature Medicine*, 2018) involving pigtail macaques, including comparing and contrasting their findings with those observed in rhesus macaques in this study. Please see lines 98-102 and 376-381.

Concern: experiments showing that a lower inoculum in pregnant NHPs results in similar pathogenesis would strengthen the case for publication with us.

Response: Our ongoing studies with 100-fold lower ZIKV doses delivered either IV&IA or subcutaneously produced fetal demise in 6/21 (28%) pregnancies, reproducing the 1/4 (25%) fetal fatality rate observed in this study. Furthermore, a manuscript entitled *Miscarriage and Stillbirth Following Maternal Zika Virus Infection in Nonhuman Primates* (by Dudley et al.; under review by *Nature Medicine*) describing a similar pattern of fetal death in ZIKV-infected nonhuman primates across primate centers in the United States also found a similar overall fetal demise rate of 25% (13 out of 50 animals). These parallel data suggest that the fetal death rate in the small cohort in this study is representative of rates in larger cohorts of rhesus macaques, which we feel justifies reporting only on the 4 RM delivered the higher dose in this manuscript. To reflect the unpublished data, we modified the revised manuscript as follows (lines 301 and following):

“Although only the GD41 fetus died in this study, ZIKV as a cause of fetal death is supported by observations of elevated rates of early fetal losses (5 out of 18 fetuses) in our ongoing studies with **first-trimester** pregnant macaques inoculated subcutaneously or via combined IV and IA routes **with a 100-fold lower inoculation dose than the one used in the current study.**”

The fetal loss observed in 28% of the pregnant animals in all our cohorts may parallel rates of miscarriage in women, typically difficult to measure but currently being estimated in large cohort studies in ZIKV endemic areas.”

Although we agree that demonstrating neuropathologic changes in parallel with ZIKV RNA in fetal brains from more individuals would further support our observation of ZIKV-mediated fetal neuropathology, unfortunately, the brains from all 5 cases of fetal demise in our ongoing studies were moderately to severely autolyzed at the time necropsy was performed, and the fetal loss occurred also early in gestation when the brain was less developed. Both of these factors precluded the same detailed histopathologic analyses as those described in the current manuscript. In our ongoing studies, the brain of 1 stillbirth (also inoculated with 100-fold less ZIKV) had evidence of ependymal loss (consistent with the findings described in the current manuscript), but given that mild tissue autolysis occurred (due to the time interval between detection of the stillbirth at night and the necropsy the following morning), this finding is less conclusive than the observations in the freshly harvested brain tissues described in the current manuscript.

Reviewers' comments:

Reviewer #2 (Remarks to the Author):

General

- The authors made considerable efforts to modify the manuscript based on previous reviews. These efforts strengthened the manuscript and clarified the data presented.

What are the major claims of the paper?

- The authors have restated their research claims with a focus on creating a pathogenesis versus transmission model and reducing their claims the model with broadly support evaluation of Zika countermeasures. The modifications regarding the claims are adequate.

Are they novel and will they be of interest to others in the community and the wider field?

- They model is exploratory and methodological challenges remain. That being said, it does add to the larger body of literature and will be interest to the wider field.

If the conclusions are not original, it would be helpful if you could provide relevant references.

- Referencing is adequate.

Is the work convincing, and if not, what further evidence would be required to strengthen the conclusions?

- Elements of CZS are evident in most animals but there is variation among the small sample size. The dual IV and IA inoculation method, despite the reasonable explanation for its use, weakens the overall model because it bypasses processes which may potentially contribute to overall pathogenesis.

On a more subjective note, do you feel that the paper will influence thinking in the field?

- Inform yes, influence no.

Please feel free to raise any further questions and concerns about the paper.

- The authors have done an adequate job addressing previous reviews to the extent the study methods allow.

We would also be grateful if you could comment on the appropriateness and validity of any statistical analysis, as well the ability of a researcher to reproduce the work, given the level of detail provided.

- The level of detail is appropriate.

We thank reviewer #2 for this positive review of the manuscript.

REVIEWERS' COMMENTS:

Reviewer #4 (Remarks to the Author):

There were prior concerns raised regarding lack of quantification of Sox and nestin in fetal brains. To address these concerns the authors used image software (Image J) to quantify both Sox2+ nuclei and Sox2 intensity staining in a region of interest (ROI). Nestin staining is more difficult to quantify because it is not nuclear and also would span different z stacks. The methodology used is therefore appropriate and the results support the conclusion that neural stem cells of the dentate gyrus are reduced in this zika model.